# Oral Electrolyte and Water Supplementation in Horses

**DOI:** 10.3390/vetsci9110626

**Published:** 2022-11-10

**Authors:** Michael Ivan Lindinger

**Affiliations:** The Nutraceutical Alliance Inc., Guelph, ON N1G 0E3, Canada; michael@nutraceuticalalliance.ca

**Keywords:** dehydration, sweating, thermoregulation, fluid balance, sodium, potassium, magnesium, calcium, transport

## Abstract

**Simple Summary:**

Heat stress results in increased sweating and respiratory rate. Sweating, especially, results in dehydration unless effective hydration strategies are employed. Effective hydration requires the supplementation of both electrolytes together with adequate amounts of water. Dehydration impairs performance, both physical and mental, and places both the horse and rider at risk of injury/death. Horses require a lot of water. Horses require a lot of electrolytes. Horses require the right balance of electrolytes in order to optimize neuromuscular function. You can lead a horse to water and it will drink; patience is required to train horses to do this. A hydrated horse is a healthy horse.

**Abstract:**

Horses that sweat for prolonged periods lose considerable amounts of water and electrolytes. Maintenance of hydration and prevention of dehydration requires that water and electrolytes are replaced. Dehydration is common in equine disciplines and can be avoided, thus promoting equine wellness, improved performance and enhanced horse and rider safety. Significant dehydration occurs through exercise or transport lasting one hour or more. Oral electrolyte supplementation is an effective strategy to replace water and electrolytes lost through sweating. The stomach and small intestine serve as a reservoir for uptake of water and electrolytes consumed 1 to 2 h prior to exercise and transport. The small intestine is the primary site of very rapid absorption of ions and water. Water and ions absorbed in the small intestine are taken up by muscles, and also transported via the blood to the skin where they serve to replace or augment the losses of water and ions in the body. Effective electrolyte supplements are designed to replace the proportions of ions lost through sweating; failure to do so can result in electrolyte imbalance. Adequate water must be consumed with electrolytes so as to maintain solution osmolality less than that of body fluids in order to promote gastric emptying and intestinal absorption. The electrolyte supplement should taste good, and horses should be trained to drink the solution voluntarily prior to and during transport, and prior to and after exercise.

## 1. Introduction

It has been known for centuries that horses lose considerably more electrolytes in their sweat than do humans [1]. This is visually evident in the amount of salt present on the hair coat (horses) or skin (humans) once the sweat evaporates. Techniques used to measure the minerals present in samples of sweat have confirmed this, the concentrations of all electrolytes (electrically charged minerals) are greater in equine sweat than human sweat (Table 1). In addition to the concentrations of electrolytes in sweat, sweating rates normalized to body surface area or to body mass are also much greater in horses than in humans (Table 1). The combination of high electrolyte concentrations and high sweating rates translates to very high rates of electrolyte loss from the body of the horse, compared to humans, during periods of heat stress. In many equine disciplines these important differences between horses and humans continue to be underappreciated. The result is that thousands of horses involved in training, competition and transport experience completely avoidable periods of moderate to severe, life-threatening dehydration and thermal strain [2,3,4,5].

Part of the problem, for the human understanding equine physiology, is that many people erroneously think that horses are not so different from humans with respect to thermoregulation and sweat production. The other part of the problem is that most horses are very tolerant of large water and electrolyte losses to the point where the losses can be life-threatening. Transport is also considered in this review because the maintenance of postural stability in a moving vehicle is actually moderate intensity exercise with sweating rates of 15 L/h or more on warm days [6,7]. Sweating rate is high when in a poorly ventilated vehicle and when the horse is wearing a cover that prevents heat dissipation. For these reasons it is recommended that transport must be well ventilated and covers used on the horse only when absolutely necessary, i.e., for protection against the cold.

The purpose of this review is to bring together the research conducted over the past 30 years illustrating the requirement for effective oral electrolyte supplementation in horses in general and athletic horses in particular. The reader is referred to two previous excellent reviews on the topic that remain highly relevant today [2,3,8,9]. The goal is to improve the health and well-being of all horses engaged in transport, training and competition. Please note that in some places it appears that a statement made by the author should be supported by a published study (reference). However, when there is no reference it is because there has not yet been a study that provides data to support it. Rather, when this occurs it is based on the author’s experiences of conducting research on more than 1000 horses at dozens of equine events over a period of more than 25 years. It is also based on owning horses and training them to freely consume water containing electrolyte supplements.

## 2. Electrolyte and Water Losses through Sweating

It is evident from Table 1 that horses have a very large engine (contracting muscles) capable of producing a lot of heat very quickly, but relative to humans have a very small radiator for dissipating metabolic heat (surface area). The horses’ BM: SA ratio is 2.5 fold greater than for humans. This translates to very rapid increases in body heat storage and core body temperature during periods of exercise and heat stress [19]. Horses try to lose this heat through the skin and through the respiratory tract [13,20]; both these routes result in loss of water from the body.

Horses lose water and electrolytes through breathing, the kidneys and the skin. Respiratory losses of water may be substantial (>10% of total water losses) during exercise in dry environments [5,20,21]. With exercise, renal losses decrease and are low. The primary route of electrolyte and water loss during periods of heat stress (exercise, transport) is the skin [2,3,13,22].

The sweating responses in horses in general and the role of the equine skin in particular are the topic of an excellent review [13]. The main practical points to keep in mind are:Sweating rates increase with increase in heat stress, whether due to ambient temperature and humidity [14,23] or intensity of exercise;Sweating rate in hot, humid conditions is higher than in hot, dry conditions due to the greater thermal stress for a given temperature arising from the decreased ability to evaporate sweat for cooling [14];High sweating rates can be sustained for more than two hours, especially when horses are adequately supplemented with effective electrolyte solutions;High sweating rates will result in dehydration when effective electrolyte supplementation is not provided;Cl^−^ is the predominant ion lost in sweat, followed by Na^+^, K^+^, Mg^2+^ and Ca^2+^;Sweat Cl^−^ losses are nearly equal to the combined losses of all cations;Sweating rates (and hence thermoregulatory cooling) decrease as horses become dehydrated.Electrolytes are required in body fluid compartments in order to retain the water in these compartments. Thus consuming only water to try to rehydrate will only dilute the body fluid compartments, resulting in renal water excretion together with more electrolytes. Water alone cannot rehydrate, and can further dehydrate [24].

## 3. Mineral Imbalance/Deficiency

Exercise-induced losses of water and electrolytes results in decreases in the content of water and electrolytes in both the intracellular and extracellular fluid compartments [25,26]. The magnitude of these losses, and the resultant changes in extracellular and intracellular ion concentrations, negatively affects nerve and muscle function [27], and therefore physical [28] and mental performance [29].

The first controlled study to report water and ion losses was performed using Thoroughbreds completing 80 km of endurance type exercise at a speed of 16 to 18 km/h [16,30]. The horses did not receive water and electrolyte supplements. The total estimated loss of water was 40 L, with decreases in total body electrolytes of 2.2 moles of Na^+^, 2.15 moles of Cl^−^ and 0.02 moles of K^+^ [31]. A detailed analysis of studies performed prior to 1995, including an additional 13 endurance rides that were the subject of study by the author, reported that horses lost 2.6% (=12 L) to 7.4% (=33 L) of their body mass (BM) as water over ride distances ranging from 50 km (2.6% of BM) to 162 km (7.4% of BM) [31]. Losses of ions similarly increased with increasing distance, with loss rates averaging about 90% of the values reported by Snow [16,30]. Losses of water and electrolytes increased in proportion to increasing difficulty of terrain, increasing ambient temperature and increasing speed of the horse [31]. In a study where horses exercised at 50% of peak VO_2_ for 45 min (~12 to 16 km based on data of Lund and Guthrie 1995 [32], cumulative loss of water was 11 L, while losses of Na^+^, K^+^ and Cl^−^ were 863, 79 and 1107 mmoles, respectively [25]. In summary, the losses of water and electrolytes are directly related to the intensity and duration of exercise resulting in thermal stress.

## 4. Dehydration

Failure to adequately replace water and ions lost through sweating will result in dehydration. The term clinical dehydration has traditionally been applied to a loss of fluid equal to or greater than 3% of body mass, i.e., 15 L for a 500 kg horse [5]. A more functional and relevant definition is the loss of body water at a rate greater than the ability to replace it [33]. Clinical dehydration is of concern whenever a horse is sweating for durations exceeding one hour, and this includes the post-exercise recovery period after high-intensity exercise such as can occur with horse race training and competition [34]. A dehydrated horse should first be administered an adequate amount of water and electrolyte supplementation prior to eating dry feeds, further exercise or transport.

In this context the role of the hindgut as a potential reservoir for fluid and electrolytes to replace sweat losses in the exercising horse needs to be considered [35]. The volume of the hindgut of a 500 kg horse is about 150 L of which the principle molecule is water, and there is an abundance of electrolytes [36]. Dehydration of the extracellular fluids in the body would be expected to result in an increase in the osmotic movement of water (with electrolytes and other small molecules such as volatile fatty acids). While this is beneficial for the extracellular fluids and cardiovascular function (see below), the net effect is a dehydration of the contents of the hindgut. This is not desirable as it predisposes to impaction colics [37] and considerable time may be needed to effectively restore hindgut hydration. Therefore, with respect to the maintenance of hydration during transport and exercise one should rely on the stomach and small intestine as the first line of water and electrolyte uptake.

## 5. Acid-Base Balance

The effects of prolonged sweating on blood acid-base balance are small, but measurable and are a reflection of the sweat losses of ions. Prolonged moderate intensity exercise in horses typically results in a mild systemic hypochloremic alkalosis (low plasma [Cl^−^] with raised plasma pH; [38,39]. The hypochloremia results from the sweat loss of Cl^−^ to required to provide electrical charge balance for the total cation loss [5,38,40,41]. The electrolyte imbalance in the body fluids may compromise neuromuscular function, performance and health.

## 6. Negative Clinical Effects of Excessive Electrolyte and Water Losses

*Dehydration.* Clinical dehydration is life-threatening and effective measures should always be taken to prevent it [33]. The manifestations of clinical dehydration include muscle weakness, increased heart rate, increased respiratory rate, excessive rectal/core temperature, increase in rectal/core temperature even at low work rates, impaired mental function, muscle fasciculations, loss of skin elasticity (skin tenting with the pinch test), impaired cardiac recovery index.

The dehydration that occurs with prolonged sweating is both extracellular and intracellular [34,42]. Initially, the majority of the dehydration occurs in the extracellular fluid compartment for two reasons. First, the onset of exercise results in a net shift of protein-poor fluid (similar to lymph fluid) into contracting muscles (intracellular and interstitial) which reduces plasma volume, blood volume and raises plasma protein concentration ([PP]) [43]. Second, the fluid that forms sweat by necessity arises from plasma [13,22]. Ultimately, the loss of fluid from plasma must be replaced, and this occurs by movement of fluid from lymphatics and other whole body extracellular fluid compartments [43], the gastrointestinal tract [40] and whole body intracellular fluid compartments [25,34]. The latter ultimately results in intracellular dehydration (water and electrolyte deficits) that will at the least compromise neuromuscular function. The benefit of the intra- to extracellular fluid shift is that the large intracellular fluid compartment (2-fold larger than the extracellular fluid compartment) ‘defends’ the extracellular fluid volume, replacing some of the extracellular fluid lost through sweating. Maintaining extracellular fluid volume and hence blood volume is important for the maintenance of cardiovascular function in the face of the demands for increased cardiac output. Loss of blood volume is associated with increased heart rate in an attempt to meet the requirement for elevated cardiac output [44]. Excessive loss of extracellular volume will result in loss of renal function and catastrophic thermal strain due to an inability to remove heat from the body.

*Cardiovascular function.* Clinical dehydration results from a loss of extracellular fluid volume including lymph volume, plasma volume and circulating blood volume [5,45]. With exercise there is an increased demand for cardiac output both to contracting muscles and to the skin. In part, the increased cardiac output at both these major sites is due to vasodilation of the tissues. It is also due to the requirement to supply contracting muscle with oxygen and nutrients, to remove heat, to remove CO_2_ and also to remove lactate and K^+^ when work rates are high [46]. A high blood flow to the skin helps to move heated blood to the periphery where it can be cooled, and also to provide fluid for the cutaneous sweating mechanism [13,22,25].

*Hyperthermia.* As dehydration progresses during periods of prolonged sweating (exercise, transport, heat exposure) the ability to cool the body through all body surface mechanisms (convection, conduction, evaporative cooling) is progressively compromised [21]. In order to maintain elevated rates of blood flow to contracting muscles and the skin when circulating volume is decreased, and consequently right ventricular filling) the frequency of cardiac contraction (heart rate) must be increased [44]. Heart rate in hyperthermic horses can be very high and even when the horse is no longer exercising heart rate does not return to normal resting levels due to the need to continue to remove heat from the body (increased cardiac output). Dehydrated horses need to be treated early and effectively by cessation of unnecessary activity, large-volume oral electrolyte supplementation [25] and/or intravenous electrolyte therapy [47] and implementation of effective cooling strategies [48]. Because dehydration is a leading cause of impaction colics, rehydration of ingesta via oral administration of fluid is recommended [37]. While intravenous fluid therapy may be used to rehydrate ingesta [47], oral therapy is recommended.

## 7. Strategies for Replacing Electrolyte and Water Losses

With the understanding that sweat fluid losses are required to support thermoregulatory cooling, one cannot and should not attempt to prevent electrolyte and water losses from occurring—sweating is required. The evidence to support this arises from the condition of anhidrosis in horses which severely compromises the ability of horses to exercise [49]. The strategy [2,5,8] then is to find a way to effectively replace electrolytes and water at the same rate as these are being lost from the body during periods of prolonged (greater than 1 h) transport and exercise or during periods of recovery of short-lasting exercise in hot conditions (Figure 1).

## 8. Requirements for an Effective Electrolyte Supplement

The main goal of effective electrolyte supplementation is to replace, in the correct proportions and amounts, the electrolytes and water lost through sweating [8,9,40]. This is necessary in order to try to maintain optimum functioning of all physiological systems. In concept, this is simple. In practice it has typically been poorly done. While there are variations in sweat ion concentrations over time, between individuals and with heat acclimation [14,15,23] the differences are adequately small such that one electrolyte supplement can be designed to suit all horses reasonably well and so prevent or treat excessive dehydration [2,8,25,40]. Some individual horses, after detailed analysis of blood and sweat, may be deemed to require additional Na^+^ or K^+^, but this is not necessary to prevent and treat dehydration.

Some people may question the need for Ca^2+^ and Mg^2+^ in the supplement. Because Ca^2+^ and Mg^2+^ are lost in sweat, and because most of the Ca^2+^ and Mg^2+^ lost in sweat are arising from the extracellular fluid compartment, losses in that compartment must be replaced from either muscle or ingestion of supplements, else electrolyte imbalances occur with attendant decrements in neuromuscular function (thumps, muscle fasciculations, etc.) as described below.

Based on the human literature [50] D-glucose (dextrose) facilitates the absorption of Na^+^ and water in the small intestine, and glucose is readily taken up by intestinal epithelial cells and used as an energy source to provide ATP. It is therefore recommended that effective electrolyte solutions contain some (up to 50% by weight) dextrose to facilitate water and electrolyte absorption and improve solution palatability.

At this point it needs to be appreciated that Himalayan sea salt, Dead Sea salt, marine salts and ‘natural’ sea minerals do not come close to approximating the concentrations of electrolytes lost in sweat (Table 2)—these types of salts are widely available commercially but need to be avoided. The best supplements are man-made, using relatively pure chemicals, in order to provide an optimum amount of each electrolyte in the most bioavailable form possible.

The composition of effective electrolyte supplements should mimic the proportion of ions present in equine sweat. Proportion is important (i.e., Perform’N Win in Table 2), and proportion is more important than the concentration of each ion in the electrolyte solutions (powdered supplement mixed into the required amount of water) so that an appropriate physiological balance between the various electrolyte species is maintained. Because equine sweat ranges from isotonic to hypertonic relative to body fluids, one might think that effective electrolyte solutions should also be isotonic to hypertonic. However, experience with horses has shown that such solutions are strongly avoided by horses—likely due to both taste and mouth feel.

Research in humans has shown that oral electrolyte supplementation is most effective with respect to gastric emptying, intestinal absorption and distribution in the body when the electrolyte solution is somewhat hypotonic [50,54]. The effectiveness of this design has been demonstrated in horses [40]. Therefore, effective equine electrolyte supplements should be designed so that the correct proportions of Na^+^, K^+^, Cl^−^, Ca^2+^ and Mg^2+^ are present (and perhaps other important trace minerals could be included but are not necessary—notably Zn, Mn and Cr) and the osmolality is not greater than 70% of normal body fluid osmolarity; an osmolality of about 200 mOsm/L is recommended and has been used successfully [25,55,56].

## 9. Biovailability

Just because one provides a mineral does not mean that it is taken up into the body. Bioavailability refers to the ability of the intestinal system to ‘absorb’ electrolytes into the blood and rest of the body. Electrolytes such as Na^+^, K^+^ and Cl^−^ have high bioavailability (up to 90% of ingested electrolyte is absorbed into the body; [57,58]. In contrast, Mg^2+^ and Ca^2+^ have low bioavailability [59], ranging from as low as 20% (i.e., magnesium oxide, calcium carbonate) to 50% (i.e., amino acid and yeast chelates). Many commercial electrolyte supplements still use very poorly absorbed Mg^2+^ and Ca^2+^ sources (oxides and carbonates)—the result is often a pronounced depletion of these two minerals from the body when the horse is engaged in long-duration exercise (endurance and eventing). An overt visual manifestation of divalent cation depletion is muscle fasciculation and muscle cramping [60,61], as well as thumps [62]. Therefore, care should be taken to select only electrolyte supplements that have the more bioavailable forms of Ca^2+^ and Mg^2+^: the citrates, the chlorides, the amino acid chelates, yeast chelates [63].

## 10. Taste and Training the Horse to Drink an Electrolyte Supplement

There is no published, scientific evidence to support the very important point that horses need to be trained to drink electrolyte supplements and how this can be accomplished. The information below comes from the author’s experience and observations over more than 25 years of scientific research.

Horses are sensitive to the taste of solid and liquid foods and this affects the ability of horses to eat and drink [64,65]. Often, presentation of liquids with novel taste and mouth feel such as an electrolyte supplement results in an initial aversion to the solution and a low or no rate of consumption. Horses can be trained to drink, and to drink electrolyte supplements. The approach I have used successfully when an initial aversion occurs is to dilute the solution 5-fold and not offer a choice. The horse may choose to not drink for an hour, or even up to 6 h. However, eventually the horse will drink the solution. After this has occurred, the strength can be gradually increased, over a period of 3 to 7 days, so that the horse eventually has no trouble drinking the solution at full strength. Additionally, horses like to engage in favorite activities, and a horse can be gently dissuaded from participation in a favorite activity until drinking a desired volume of electrolyte solution. The keys to success are persistence, no choice and gentle encouragement. The horse will gradually (over a period of one to 7 days) become used to both drinking, and to drinking something with a new taste (electrolyte solution).

## 11. Pastes and Slurries

In large part because of aversion of horses to voluntarily drinking effective electrolyte solutions the approach has been taken to try to provide adequate amounts of electrolytes via syringe into the back of the mouth using pastes and slurries [56,66,67,68]. While it is possible to give adequate amounts of electrolytes this way, it is very typically not accompanied by adequate amounts of water. This can result in a very concentrated salt solution in the stomach, which is retained in the stomach (high osmolarity inhibits gastric emptying [69,70]), irritates the stomach lining (contributing to gastric lesions [71]) which slowly leaves the stomach. Additionally, due to the high osmolarity of the resultant stomach and small intestine contents, there can be an osmotic movement of water from body fluids into the stomach and small intestine. This is water moving in the opposite direction from what is desired and can make an already dehydrated horse more dehydrated. If concentrated solutions must be used, then it is imperative that adequate amounts of water are consumed in very soon after so that solution osmolarity in the stomach and small intestine does not exceed an osmolarity of 280 mOsm/L.

## 12. Electrolytes Top Dressed on Feed

This commentary is based on observation and experience. It is common practice to top dress electrolytes on to the feed as one way of delivering electrolytes. This is not an effective strategy for replacing electrolytes and water lost during exercise or transport that has resulted in significant dehydration. Hydration should be achieved first, prior to allowing the horse to eat dry food stuffs. An alternative of this, used successfully by many endurance teams both during the ride and after completion, is to place food stuffs, with electrolytes and an adequate volume of water into a tub, allowing the dry food stuffs to absorb water. Using this approach it is possible to deliver adequate amounts of water and electrolytes over the duration of the ride and initial period of recovery to replace sweat losses.

## 13. Distribution of Ingested Electrolytes in the Body after ‘Absorption’ and Appearance in Sweat

Using radioactive imaging techniques we were able to determine the rate of gastric emptying of 3 L of an electrolyte supplement nasogastrically delivered into the stomach [40]. The electrolyte supplement was Perform’N Win (Buckeye Nutrition, Dalton, OH, USA) with the composition provided in Table 2. The half-time of gastric emptying was 47 min, suggesting that the stomach effectively served as a reservoir capable of replenishing water and electrolyte over a two hour time frame. This is important for both exercise and transport activities. In human studies, larger volumes of ingested solutions empty with a similar time frame [72], i.e., the half-time for gastric emptying in horses will be approximately 45–50 min. Thus when 8 L of electrolyte supplement is orally ingested in a short period of time, up to 5 L of this solution can empty from the stomach and pass into the small intestine for absorption within a one-hour period.

Using radiotracer techniques for ingested Na^+^ and K^+^ we have been able to trace the uptake of Na^+^ and K^+^ consumed with an electrolyte supplement into the blood, and also to the muscle [25] and its appearance in sweat during exercise [22]. When 3 L of electrolyte supplement was administered into the stomach, ingested Na^+^ and K^+^ appeared in the plasma within 10 min of administration, with peak plasma values occurring at 20 to 40 min after ingestion [25,40]. Plasma concentrations of radiotracer Na^+^ and K^+^ remained at peak values for up to 2 h post-administration, indicating a steady rate of delivery from the stomach and absorption by the small intestine. Plasma values remained elevated for 20 h, indicating ongoing exchange of radiotracer ions with both intracellular and extracellular ion pools. From these data it can be concluded that the stomach does effectively serve as a reservoir to replace water and ions lost in sweat over at least a 2 h period of time. While the studies have yet to be performed, it is suggested that drinking additional electrolyte supplement at 15 to 30 min intervals, thus increasing gastric volume, will accelerate the amount of fluid emptied from the stomach and absorbed by the small intestine.

It is important that ingested ions are getting to where they are needed in the body. Most of the Na^+^ and Cl^−^ appearing in sweat comes from the extracellular fluid compartment, while most of the K^+^ and some of the Cl^−^ arises from the intracellular fluid compartment. We know this because the amount of Na^+^ lost can approach the total intracellular Na^+^ content and that the amount of K^+^ lost can easily exceed the total extracellular K^+^ content [22]. K^+^ is the predominant cation present in muscle and Cl^−^ is the predominant anion, with both contributing to muscle osmolality and fluid volume. Thus, K^+^ lost from muscle (and nerves) needs to be replaced for both restoration of intracellular [K^+^] and muscle intracellular volume if muscle function is to be maintained at high levels [27]. We have shown, using radiotracer technique, that ingested K^+^ is measurably present within skeletal muscle cells within 1 h [25]. Therefore, K^+^ ingested as part of electrolyte supplements is getting to where it is needed, replacing K^+^ within cells. In contrast, Na^+^ is primarily being retained within the extracellular fluid compartment, where is it needed.

An indication that the ingested radiotracers of Na^+^ and K^+^ are behaving in a manner similar to their non-radioactive counterparts is the presence of both radiotracers in sweat within the first 10 min of exercise. The radiotracers reached peak values within 30 min of exercise and remain elevated through the exercise period into early recovery—until cessation of sweating [22]. From these data it can be concluded that ingested electrolytes and water are being used, in part, to produce sweat and, in part, to replace water and ions that may have been lost from extra- and intracellular fluid compartments. The contribution of ingested water and ions directly to sweat formation thus serves to mitigate losses of water and ions that would have had to have come directly from extra- and intracellular fluid compartments.

## 14. What Happens If Too Much Electrolytes Are Given?

If excessive amounts of electrolytes have been provided together with adequate amounts of water there will be no negative effects on horse health. The extra electrolytes will be excreted by the kidneys as part of normal kidney function. This of course presumes normal renal function of the horse, a situation that may not be present in a severely dehydration horse [73,74].

If, however, not enough water has been provided, then the electrolytes may sit in stomach and act as both a desiccant and an irritant [71]. The high osmolality in the stomach and duodenum will draw water from the blood, further dehydrating body fluid compartments. Only after duodenal osmolality becomes less than that of the body fluids will water and electrolytes move in the correct direction, from gastrointestinal tract back into blood/cells.

The message here is that, similar to pastes and slurries, just because one administers electrolytes to the horses does not mean that the electrolytes are getting into the horse. Adequate amounts of water are required.

## 15. When to Administer—Timing of Electrolyte Supplementation

This commentary is based on observation and experience. In general, non-athletic horses do not need to be provided with electrolyte supplementation. There is a wide variety of methods people have used to try to provide adequate amounts of electrolytes to their horses. These include providing electrolytes at mealtimes, after exercise and training, sometimes before exercise and training. Ideally, one wants to use preventative strategies as opposed to treatment strategies. When this consideration is embraced, then it becomes clear that electrolyte supplementation should be provided about one hour before transport/exercise. This is an effective preventative strategy because the ingested water and electrolytes replaces losses in real time. When exercise is continued beyond one hour, then supplementation should be provided as needed to prevent dehydration, so that losses are replaced in real time. Post exercise and post transport supplementation is also important to try to fully replenish and rehydrate—this should be done prior to ingestion of dry food stuffs.

The research conducted to date provides a good indication of when electrolyte supplementation is most effective. This varies somewhat by discipline and the recommendations are based on previous work summarized by Kronfeld [8,9] and from the author’s scientific experience.

*Short duration exercise.* When the horse is dehydrated from prior transport (30 min or longer on warm days) or had no access to water for more than one hour, then electrolyte supplementation should be administered one hours prior to even short-term exercise. This will help to ensure adequate hydration with the aim of achieving an optimum of physical and mental wellness for the activity. Regarding the practice of preventing drinking by horses competing in horseraces, there is no scientific evidence that this is a benefit to speed and performance despite the loss of weight associated with this form intentional (on the part of the horse person) dehydration. In general, both mental (cognitive) and physical functions are impaired by even mild dehydration.

*Transport.* Because transport always results in sweating, even on cool days, it should be considered best practice to administer electrolyte supplementation during the hour preceding loading. This helps to ensure that sweat losses occurring during transport are offset by water and electrolyte absorption from the gastrointesintal tract into the body. When this is done well, the horse will arrive up to 2 h after loading with minimal or no dehydration. If transport exceeds two hours, then rest breaks with administration of electrolyte supplements should be provided. In cases where breaks are not feasible at 2 h intervals, the transport box should be equipped with a watering device that provides the electrolyte supplement, and not water.

*After exercise.* Many people think that once the training and the competition is over, there may be little ongoing need for electrolyte supplementation. However, horses participating in short-term high intensity exercise may sweat for an hour or more following cessation of the activity, resulting in up to 15 L of fluid loss. This is a significant dehydration and one that should be avoided. Furthermore, the post-exercise dehydration should be corrected prior to transport of the horse because transport will further dehydrate the already dehydrated horse.

*Long duration exercise.* Long duration exercise can be considered any form of activity where there are intentional rest periods during the activity, such as in the cross country phase of eventing and endurance rides. Since the difficulties associated with overheated horses at the 1992 summer Olympic games in Barcelona, increasing attention has been given to the work intensity and effective strategies for cooling and hydration [75,76]. The rest interval in long forms of the cross country now provide an opportunity for cooling as well as water and electrolyte replacement. Post-cross country rehydration is crucial when jumping is to be performed the next day and sometimes even later in the same day as the cross country trial was completed.

By way of example, it is possible to complete a 100-mile endurance ride with no detectable dehydration at any of the veterinary checkpoints. I personally observed this in the 1995 Race of Champions by the horse ‘Cash’ then owned and ridden by Valerie Kanavy. The key points are these: ensure hydration prior to starting the ride/race by using electrolyte supplementation—water alone cannot result in effective hydration. Take every opportunity to provide electrolyte supplementation *and ensuring the drinking of adequate amounts of water* during the ride: every rest break, whenever there is access to potable water (streams, lakes, ponds) and providing work breaks for the horse (i.e., jogging along side the horse) when the going is difficult.

## 16. Conclusions

The losses of water and electrolytes are directly related to the intensity and duration of exercise resulting in thermal stress. The principle anion Cl^−^ is the predominant ion lost in sweat, followed by Na^+^, K^+^, Mg^2+^ and Ca^2+^. The inability to match the rate of water and electrolyte losses occurring through sweating will result in dehydration. A dehydrated horse should first be administered an adequate amount of water and electrolyte supplementation prior to eating dry feeds, further exercise or transport. The main goal of effective electrolyte supplementation is to replace, in the correct proportions and amounts, the electrolytes and water lost through sweating. The best supplements are man-made, using relatively pure chemicals, in order to pro-vide an optimum amount of each electrolyte in the most bioavailable form possible. Electrolyte supplementation should be provided about one hour before transport/exercise. This is an effective preventative strategy because the ingested water and electrolytes replaces losses in real time.

## Figures and Tables

**Figure 1 vetsci-09-00626-f001:**
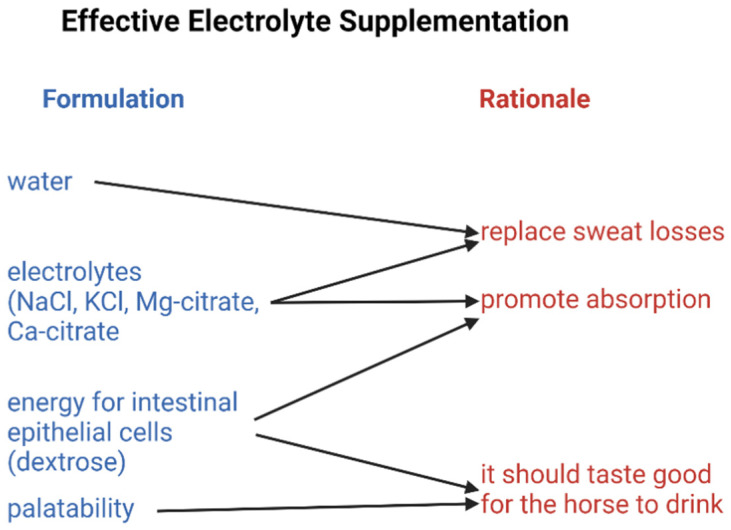
Conceptual diagram illustrating requirements for effective water and electrolyte replacement. Created using Biorender.

**Table 1 vetsci-09-00626-t001:** Horse versus human sweating and evaporative cooling.

Parameter	Horse	Human
Body mass—BM (kg)	500	80
Contracting muscle (kg) ^a^	200	16
Surface area (SA) for cooling (m^2^) ^b^	5.09	1.8
BM: SA ratio	100	40
Sweating rate (mL·m^−2^·min^−1^) ^c^	50	30
% sweat used for cooling ^d^	25–30	30–50
Total sweat [ion]s] (mmol/L) ^e^	200	50
Sweat [Na^+^] (mmol/L) ^e^	120	40
Sweat [K^+^] (mmol/L) ^e^	60	4
Sweat [Cl^−^] (mmol/L) ^e^	180	50
Sweat [Ca^2+^] (mmol/L) ^f^	3 to 7	40
Sweat [Mg^2+^] (mmol/L) ^f^	3 to 6	4
Sweating rate (L/h) ^e^	10–15	2–3

^a^ Gunn 1987 [10]; ^b^ Hodgson et al. 1994 [11]; ^c^ Kingston et al. 1997 [12]; ^d^ Jenkinson et al. 2006 [13]; ^e^ McCutcheon et al. 1995, 1999 [14,15]. ^f^ Kerr and Snow 1983; Carlson and Ocen 1979; McConaghy et al. 1995 [16,17,18].

**Table 2 vetsci-09-00626-t002:** Composition of different types of salts commonly used in equine salt supplements, compared to the composition of horse sweat.

Source	NaCl	Ca	Mg	K
	% Dry Weight			
Redmond–Rock Crushed Loose Mineral Salt for Horses ^a^	91	0.35	0.06	0.03
Himalayan Pink Salt ^b^	50–70	0.18–0.31	0.13–0.25	0.21–0.35
Dead Sea Salt ^c^	30	7	25	2
Ocean salt	97			
Perform’N Win ^d^	33	0.024	0.03	18
Equine Sweat ^e^	48	0.04	0.05	14

NaCl, sodium chloride; Ca, calcium; P, phosphorous; Mg, magnesium, K, potassium. The principle anion associated with the cation minerals is chloride (Cl). ^a^ https://redmondequine.com/redmond-rock-crushed (accessed on 7 October 2022); ^b^ (Fayet-Morre et al. 2020 [51]; ^c^ Porath et al. 1989 [52]; ^d^ Lindinger and Waller 2022; Perform’N Win is nearly 50% by weight of dextrose [53]; ^e^ Kerr and Snow 1983; McCutcheon et al. 1999 [15,16].

## Data Availability

No new data are presented in this review article. The articles cited are the sources of the data.

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
