# Peer review of "Oral Electrolyte and Water Supplementation in Horses"

_vetsci, 2022, doi:10.3390/vetsci9110626_

Round 1

Reviewer 1 Report

Please see attached document for specific points.  The author of this review appears very knowledgeable but does not really present a comprehensive review of the literature.

Reviewer 2 Report

Review of “Oral Electrolyte and Water Supplementation in Horses”

Summary: The topic of water and electrolyte supplementation for equine athletes is a critical topic. An up-to-date literature review on the topic would be of significant value. In its current form, this article falls short of that need. Dr. Lindinger is a leader and respected scientist in this area and capable of providing a valuable contribution in this area. In its current form, this article seems rushed with a lack of detailed information, lack of sufficient references in several areas, and grammatical errors through the paper.

Specific comments:

Page 1, lines 16-17; Can you be more precise regarding recommendations for adequate water here and throughout the manuscript?

Page 1, line 19; The recommendations for electrolyte and water supplementation to horses during exercise are limited here and throughout. Adding more in this area would strengthen this review article.

Page 1, lines 40-41; Your point here is a bit unclear. Could this be stated more clearly? This indicates that somehow horses are more tolerant than humans to large water and electrolyte losses… until they die? While there may be some truth here, I think that this could be stated more clearly or completely for this review.

Page 2, Table 1; Please provide a reference for where these values originated.

Page 2, lines 65-81; It appears that perhaps there was supposed to be some sort of bulleted list here?

Page 2, lines 72-73; Should probably say “…. When effective water and electrolyte supplementation is not provided.”

Page 3, line 86; Here (“negatively”) and throughout the manuscript are several minor grammatical or awkward wording mistakes. Care should be taken to carefully read through to find all of these.

Page 3, lines 128-129; awkward wording.

Page 4, lines 157-159; Can a bit more explanation regarding how intravenous fluid therapy compromises gut function be provided?

Page 4, line 161; awkward wording.

Page 5, lines 177-181; While the statements made here are correct, a clearer presentation of the evidence would strengthen the point. For example a table that presents the concentrations of electrolytes in some of these examples.

Page 8, lines 349-351; This last sentence seems an odd end to a review on water and electrolyte supplementation. It is too informal and off topic. In fact the sentence does not even communicate its point clearly.

Page 10, line 450; I do not believe there is any 43rd reference in the manuscript.

General Comments:

1.       There are several sections that seem under-developed for a review article on this topic. Examples would include Dehydration, Acid-base balance, Pastes and slurries, What happens if too much electrolytes are given, as well as others. Not only are some of these sections under-developed, but they also lack references. As a review article, this should not simply be an opinion piece by the author but provide an adequate review of the scientific literature on the topic and sub-topics.

2.       The sub-section on “Cardiovascular function” should have been about the negative effects of excessive electrolyte and water losses on this function, but this does not seem to be the clear topic of that paragraph.

3.       In reviewing this article, I re-read the 3 articles written by David Kronfeld on “Body Fluids and Exercise” published in the Journal of Equine Veterinary Science in 2001. It is of note that these are not mentioned in this review of the literature. While there may be some differences in interpretation, some inclusion of the data presented in those articles would strengthen this review. It is particularly noteworthy that this article does not delve appreciably into electrolyte supplementation during exercise. Highlighting differences and developments that may have occurred between 2001 and 2022 is valuable to healthy scientific discourse. Much of this is particularly relevant in section 8 of this paper. Some discussion of how things change over time is missing from the current manuscript. Obviously, the timeframe is different depending on the discipline.

4.       The section on taste and training horses to drink seems to be a bit more of the author’s experience and less a review of the science in this area. This is not to say it is unnecessary, but here and in several other places the paper could be strengthened by a greater inclusion of references.

5.       It seems that as the paper progresses, the references become less frequent. This is evident in sections such as “What happens if too much electrolytes are given?”. This lack of supporting evidence detracts from the strength of this review.

Round 2

Reviewer 2 Report

Thank you for making significant changes to the original draft of this manuscript. I remain concerned regarding the publication of "experience" as a criteria via which to weigh this paper. There are many "experts" with years of experience whom have worked with 1000s of horses, but science and the scientific method have an important place as a keystone in the solid framework of the knowledge we use. I know that this author is the expert he claims to be, but I could be wrong. I hope he will consider the precedent that he is a part of setting.